# Research on the Influence of Inlet Velocity on Micron Particles Aggregation during Membrane Filtration

**Peifeng Lin [1],\*, Qing Wang [1], Xiaojie Xu [1], Zuchao Zhu [1], Qiangmin Ding [2] and Biaohua Cai [3]**

[1]  Key Laboratory of Fluid Transmission Technology of Zhejiang Province, Zhejiang Sci-Tech University, Hangzhou 310018, China

[2]  Hefei General Machinery Research Institute, Hefei 230032, China

[3]  Wuhan Second Ship Design and Research Institute, Wuhan 430064, China

\*  Correspondence: linpf@zstu.edu.cn

**Abstract:** Membrane filtration is an efficient wastewater treatment technology. However, sludge particles will easily aggregate and deposit upon the membrane surface, which will decrease the water productivity of membrane filaments. Focusing on the influence of velocity on particle behavior, experimental and numerical research was carried out. The $k-\varepsilon$ turbulent model, porous media model and DPM model were adopted in the simulation. The flow characteristics including pressure, velocity and particle concentration contour are discussed using different inlet velocities of 0.6, 0.8, 1 m/s. The effects of gravity were also investigated. The final evaluation suggests the best working conditions in three scenarios, which could help to suppress membrane pollution. The results indicate that when the inlet velocity is about 1 m/s, particle deposition is weakest, resulting in better water productivity.

**Keywords:** membrane filtration; membrane fouling; numerical simulation; DPM; water productivity

## 1. Introduction

Membrane filtration is widely used in water treatment, liquid–solid separation and in other fields of application. However, during the membrane filtration process, due to the influence of dynamics and chemical factors, the sludge particles will deposit, or be adsorbed, onto the membrane surface. This leads to membrane filament fouling, reducing both filtration efficiency and the membrane lifetime. With the development of computer technology, Computational Fluid Dynamics (CFD) simulation can effectively help us understand the influence of various factors in the particle aggregation process.

The factors affecting the water productivity of membrane bioreactors have been numerically and experimentally investigated by many scholars, for decades. Ling, Q et al. [1] studied the effects of modified fly ash and powdered activated carbon on sludge mixture characteristics and the operation cycle in a membrane bioreactor (MBR). The results showed that the addition of powdered activated carbon is beneficial in delaying membrane fouling and improves filtration efficiency due to the longer operation cycle of powdered activated carbon. Ou yang, K et al. [2] studied the effect of adding powdered activated carbon on the characteristics of the sludge mixture and membrane fouling in a long−running membrane bioreactor, and analyzed the mechanism of its effect on membrane fouling. The results showed that adding powdered activated carbon increases the average particle size of the flocs and decreases the viscosity of the sludge, but the effect on the sludge content is not obvious. Feng, Q et al. [3] studied the effect of flow shear force on the settling characteristics of activated sludge through the sequencing batch reactor activated sludge process (SBR) reactor model, compared the degree of change in the activated sludge floc morphology of different water flow shear forces, and analyzed the effect of flow shear force on activated sludge flocs. The results showed that the flow shear

force changed the sedimentation characteristics of the activated sludge in the SBR reactor to a certain extent due to the shear force affecting the microbial ecosystem of activated sludge, thereby changing the structure of the flocs. Karimi, H et al. [4] synthesized a reverse osmosis polyamide membrane by designing a thermal curing method in a temperature-controlled steam and water environment, which enhanced the hydrophilicity of the membrane surface and increased the permeation flux. Chang, Q et al. [5] used nano−titanium dioxide coatings to improve the hydrophilic properties of commercial ceramic microfiltration membranes. Wei, P et al. [6] periodically introduced large bubbles to reduce the pollution of the membrane surface by using the method of VOF, which effectively increased membrane water productivity. Radu, A.I. et al. [7] studied the effect of the geometry, position and cross flow velocity of the separators between the particles on particle deposition patterns to reduce the pollution on the surface of membrane filaments, which increased the water productivity of the membrane surface. Xie, F et al. [8] used Fourier transform infrared spectroscopy, scanning electron microscopy and other methods to study the effect of microporous corrugated microchannel turbulence promoters (MCTP−MPs) on the pollution characteristics of underwater flat-membrane bioreactors. The results showed that the SMBR filter cake layer thickness using MCTP−MPs is smaller, the content of organic and inorganic contaminants is lower, and the filter cake layer is more easily removed by hydraulic conditions, thus reducing membrane fouling. Liu, X et al. [9] used the method of CFD to simulate and study the influence of fixed and moving membrane filaments on the flux of membrane filaments. An increase in the pressure drop on the luminal side leads to an increase in transmembrane pressure and membrane flux. Yu, W et al. [10] discussed the impact of aluminium polychlorid (PAC) on membrane fouling and found that the addition of low-dose (PAC) could decrease membrane fouling, which is more conducive to the removal of dissolved organic matter. Yu, W et al. [11] studied the effect of Fe/Mn oxidation on the fouling of the filter cake layer and membrane pores to decrease membrane fouling. The results showed that Fe/Mn reduced the amount of two types of fouling substances in the filter cake layer and membrane pores, and increased the membrane runtime. Qaisrani, T.M. et al. [12] explored the effect of air bubbles and backwashing in decreasing membrane fouling and improving membrane cleaning efficiency through experimental methods. Bai, H et al. [13] used the thermally induced phase separation method of polyvinylidene fluoride to compare and analyze the membrane performance of different effective membrane filament lengths under different operating parameters, and found that the shorter the effective lengths of membrane filaments, the less likely that membrane fouling occurred and the higher the membrane water productivity. Han, X et al. [14] prepared a BUT−172 modified polyamide composite forward osmosis membrane to improve the fouling resistance of the membrane filament and increase the membrane water productivity. Vera, L et al. [15] investigated the effect of flushing conditions on membrane fouling and showed that gas jet−assisted backwashing improved membrane cleaning efficiency. Lu, X et al. [16] studied the effect of colloidal particles and soluble polymers on membrane fouling, and the results showed that the decrease in particle size and the increase in the adhesion of gel-like flocs, due to the secretion of hydrophobic protein biopolymers, accelerated the fouling. Deposition and cake layer formation resulted in a better mitigation of membrane fouling. Bouhabila, E.H. et al. [17] used aluminum chloride to prepare ultrafiltration nanocomposite membranes to increase the pores on the membrane surface, which increased the membrane of water productivity. Tay, J.H. et al. [18] studied the effect of shear stress generated by upstream aeration on aerobic granulation, and the results showed that appropriate shear stress is beneficial in the generation of a stable particle structure. Hong, S.H. et al. [19] studied the effect of sequencing changes in dissolved oxygen (DO) concentration on membrane permeability of underwater membrane bioreactors (MBR), and the results show that the rate of increase in transmembrane pressure (TMP) in the anoxic phase is always steeper than that in the aerobic phase, indicating that the rate of fouling in the anoxic phase is higher than that in

the aerobic phase. Lin, W et al. [20] used the computational fluid dynamics coupling method and the response surface method to compare the differences in the hydraulic performance of the full−effect membrane flow channel and the traditional inlet flow channel, and discussed the influence of the geometric parameters of the inlet flow channel and the inlet velocity on the hydraulic performance. The results showed that, compared with the traditional membrane flow channel, the pressure drop difference between the inlet and outlet, the average velocity, and the shear force, was greatly improved, and the concentration polarization efficiency and membrane fouling rate was effectively reduced. Momenifar, M et al. [21], by using the method of simulation, explored the influence of the Taylor Reynolds number and Froude number on the gravitational acceleration of particles, and found that gravity drastically decreased the clustering of bidisperse particles, whereas it could increase the clustering of monodisperse particles. Ouellette, N et al. [22] studied the influence of tracer particles on the diffusion rate in strongly turbulent water flow in the laboratory, and also compared measurements of this turbulent-relative dispersion with the longstanding work of Richardson and Batchelor, finding excellent agreement with Batchelor's predictions. Biferale, L et al. [23] compared the experimental measurement and simulation data of the Lagrangian velocity structure function in turbulent flow, and resolved an apparent disagreement between observed experimental and numerical scaling in order to generate interest and awareness amongst other researchers so that they could consider these phenomena in their studies as well.

In this paper, a simplified membrane bioreactor model is established and then simulated. The influence of different inlet velocities on particle distribution, velocity and pressure, are studied in the flow. The results of flow evaluation and particle concentration under different conditions could help develop new strategies to control membrane fouling and thus improve water productivity.

## 2. Materials and Methods

### 2.1. Geometric Model

A simplified membrane cell with a circular tube was built, with reference to the experimental model. The structure of each zone is displayed in Figure 1. The entire flow domain was divided into three zones: external flow zone with inlet and outlet nozzles, porous media zone and internal flow zone. The raw water with particles flowed into the pipe at the inlet nozzle of the external zone, partly permeating across the semipermeable membrane (modeled as porous media) because of the negative pressure at outlet1 and outlet2 of internal flow zone, and the remaining water flowed out through the outlet nozzle. The overall length of the entire pipeline was set to 500 mm, the innermost suction pipe radius was set to 2 mm, the thickness of the porous media semipermeable membrane is set to 2 mm, the thickness of the outermost outlet pipe is set to 16 mm, and the length and radius of the inlet and outlet nozzles were set to 10 mm and 2 mm, respectively.

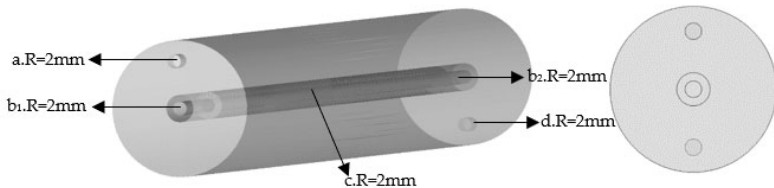

**Figure 1.** The structure diagram of each zone of the flow channel of the membrane pool, a: inlet; b1: outlet1; b2: outlet2 c: porous media semipermeable membrane; d: outlet.

*2.2. Governing Equations and Parameters*

2.2.1. Governing Equation

The flow was controlled by the continuity equation and momentum equation:

$$\frac{\partial \rho}{\partial t} + \frac{\partial}{\partial x_i}(\rho u_i) = 0 \tag{1}$$

$$\frac{\partial}{\partial t}(\rho u_i) + \frac{\partial}{\partial x_j}(\rho u_i u_j) = -\frac{\partial p}{\partial x_i} + \frac{\partial}{\partial x_j}\left[\mu\left(\frac{\partial u_i}{\partial x_j} + \frac{\partial u_j}{\partial x_j}\right)\right] + \frac{\partial}{\partial x_j}\left(-\rho \overline{u_i' u_j'}\right) + S_i \tag{2}$$

where $S_i$ is the addition of a momentum source term.

The turbulent *k-ε* model was used to take turbulence into account.

$$\rho \frac{\mathrm{d}k}{\mathrm{d}t} = \frac{\partial}{\partial x_i}\left[\left(\mu + \frac{\mu_\mathrm{t}}{\sigma_\mathrm{k}}\right)\frac{\partial k}{\partial x_i}\right] + G_\mathrm{k} + G_\mathrm{b} - \rho\varepsilon - Y_\mathrm{M} \tag{3}$$

$$\rho \frac{\mathrm{d}\varepsilon}{\mathrm{d}t} = \frac{\partial}{\partial x_i}\left[\left(\mu + \frac{\mu_\mathrm{t}}{\sigma_\mathrm{k}}\right)\frac{\partial \varepsilon}{\partial x_i}\right] + C_{1\varepsilon}\frac{\varepsilon}{k}(G_\mathrm{k} + C_{3\varepsilon}G_\mathrm{b}) - C_{2\varepsilon}\rho\frac{\varepsilon^2}{k} \tag{4}$$

where $\mu$ is dynamic viscosity, $\mu_t$ is the turbulent viscosity coefficient, $G_k$ is the turbulent kinetic energy generation due to the mean velocity gradient, $G_b$ is the turbulent kinetic energy generation with floating effects, $e$ is the turbulent dissipation rate, $k$ is the turbulent kinetic energy, $Y_M$ is the influence of compressible turbulent pulsating expansion on total dissipation rate, and $C_{1\varepsilon}$, $C_{2\varepsilon}$, $C_{3\varepsilon}$ are model parameters.

The standard wall function, based on Launder, B.E. and Spalding, D.B. [24], was used in this numerical calculation. Furthermore, by setting the boundary layer close to the wall, the mesh near the wall was refined, and the y+ of the first layer near the wall was set to about 30.

The particle's motion is controlled by the equation:

$$\frac{d\vec{u}_p}{dt} = \frac{\vec{u} - \vec{u}_p}{\tau_r} + \frac{\vec{g}(\rho_p - \rho)}{\rho_p} + \vec{F} \tag{5}$$

$$\tau_r = \frac{\rho_p d_p}{18\mu}\frac{24}{C_d Re} \tag{6}$$

$$Re \equiv \frac{\rho d_p \left|\vec{u}_p - \vec{u}\right|}{\mu} \tag{7}$$

here $\tau_r$ is the particle relaxation time, $\vec{u}$ is the fluid phase velocity, $\vec{u}_p$ is the particle velocity, $\mu$ is the molecular viscosity of the fluid, $\rho$ is the fluid density, $\rho_p$ is the density of the particle, $d_p$ is the particle diameter, $Re$ is the relative Reynolds number, and $\vec{F}$ is the virtual quality force.

Darcy's law describes the linear relationship between the seepage velocity and hydraulic gradient of water in saturated soil, also known as the linear seepage law. Porous media were modeled by the addition of a momentum source term to the standard fluid flow equations. The source term is composed of two parts: a viscous loss term and an inertial loss term.

$$\nabla p = -\frac{\mu}{\alpha}\vec{v} \tag{8}$$

$$\nabla p = -\sum_{j=1}^{3} C_{2_{ij}}\left(\frac{1}{2}\rho v_j |v|\right) \tag{9}$$

where $C_2$ is the inertia resistance coefficient, and $\frac{1}{\alpha}$ is the viscous resistance coefficient. The parameters $\frac{1}{\alpha}$ and $C_2$ are determined by experimental results, as shown later.

### 2.2.2. Parameters

The coefficient of viscous resistance and the coefficient of inertial resistance of the porous media was obtained by water productivity experiments at different pressures. The experimental equipment included a tube membrane battery, membrane wire, a pressure gauge, a fixed membrane plug, a 250 mL beaker and a circulating water pump, as shown in Figure 2. After fixing the membrane filament with a perforated cover plate, the membrane filament was placed into the circular tube membrane pool, the membrane pool was filled 90% with water so that all the membrane filaments are submerged in water, and the various pressure gauges were adjusted to show pressures of 0.04, 0.05, 0.06, 0.07, 0.08, 0.09, 0.10, and 0.11 MPa. The membrane filament produced water productivity at the different pressures. To minimize errors, the experiment was repeated. The averaged experimental results are shown in Table 1.

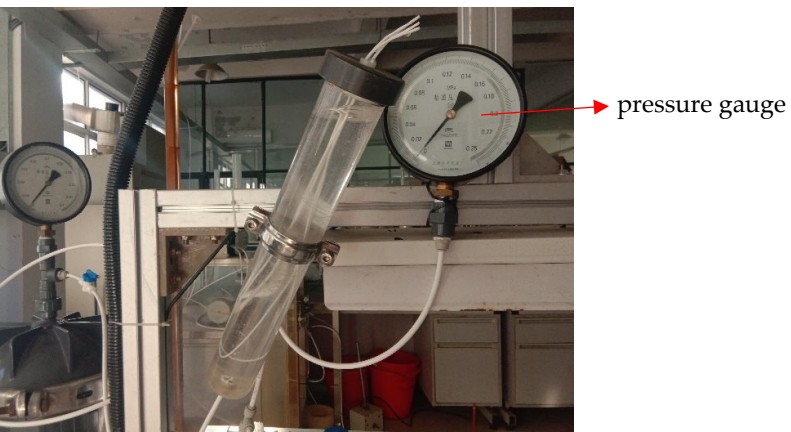

**Figure 2.** Illustration of experimental setup.

**Table 1.** Experimental water productivity and velocity at different pressures.

| P/MPa | Q/(mL/min) | v/(m/s) |
|---|---|---|
| 0.04 | 78.5 | 0.11 |
| 0.05 | 87.9 | 0.14 |
| 0.06 | 100.9 | 0.17 |
| 0.07 | 111.2 | 0.21 |
| 0.08 | 129.3 | 0.24 |
| 0.09 | 150.5 | 0.29 |
| 0.10 | 168.1 | 0.32 |
| 0.11 | 175.3 | 0.41 |

Finally, as shown in Figure 3 the viscous resistance coefficient and the inertial resistance coefficient were obtained using the fitting calculation, and were found to be $2.05 \times 10^5$ and 332, respectively.

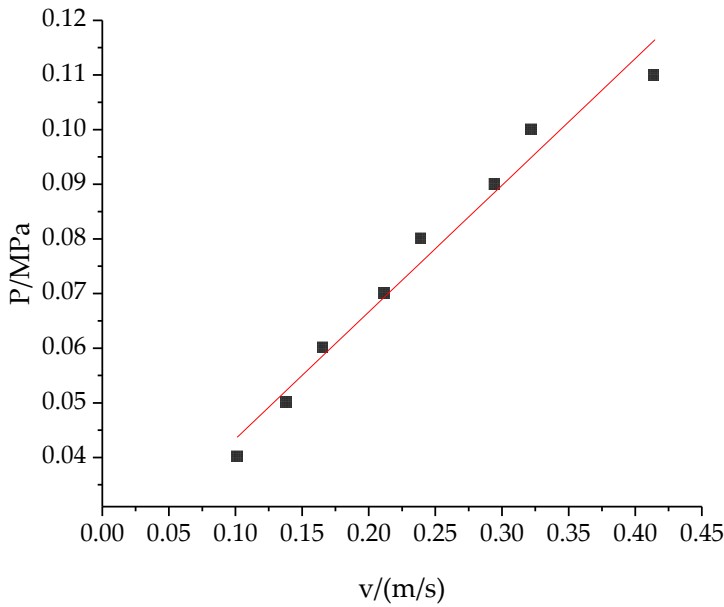

**Figure 3.** Fitting Curve of experimental value.

### 2.3. Numerical Methods and Boundary Conditions

Each zone is divided into a structured grid to improve the quality of the grid. As shown in Figure 4, the grids of the porous media zone and the boundary layer zone are respectively encrypted, in order to improve the calculation accuracy of the core zone. The overall grid number is about 500,000; the grid quality is about 0.8. The skewness of the mesh is between approximately 0.3 and 0.4. A boundary layer is added near each wall of the porous media, and the y + =30.

ANSYS FLUENT was used for numerical simulation, and a pressure−based solver was used. The convection term of the equation adopts the second-order upwind discrete format, and the other terms of the equation adopt the central difference format. The SIMPLE algorithm was used to separate and iteratively solve the velocity and pressure. Turbulence intensity was set to 5%. If the relative residual value of each variable was less than $10^{-5}$, it was considered that the results had converged.

The inlet velocity condition at which the particle aggregation effects were to be investigated, was set. The outlet pressures at outlet1 and outlet2 were both set as −5 Pa. The walls were all set as non−slip walls.

In the discrete phase setup, the particles were set to be injected at the inlet surface with mass flow rates for the three operating conditions of 0.013188 kg/s, 0.017584 kg/s and 0.02198 kg/s, respectively. The particle boundary conditions were the escape conditions of the inlet and outlet. The surface of the porous media was set as the trap condition.

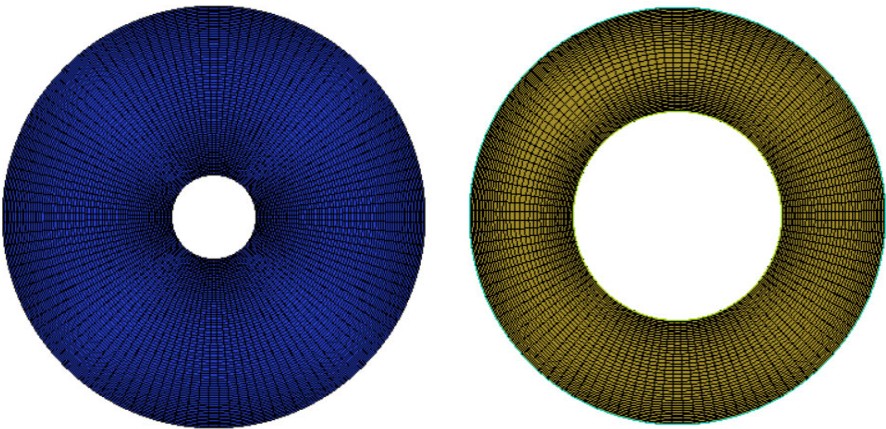

**Figure 4.** Schematic diagram of boundary layer and porous media zone mesh refinement.

*2.4. The Influence of the Grid on the Calculation Results*

Three simulation results based on grids of 300,000, 500,000, and 800,000 were compared, in order to verify the independence of the grid. Figure 5 shows the curve of water productivity rising with time. It can be seen from the curve that water productivity was basically the same under these three grids. Therefore, grid independence was verified. In this study, 500,000 grids were used for simulation.

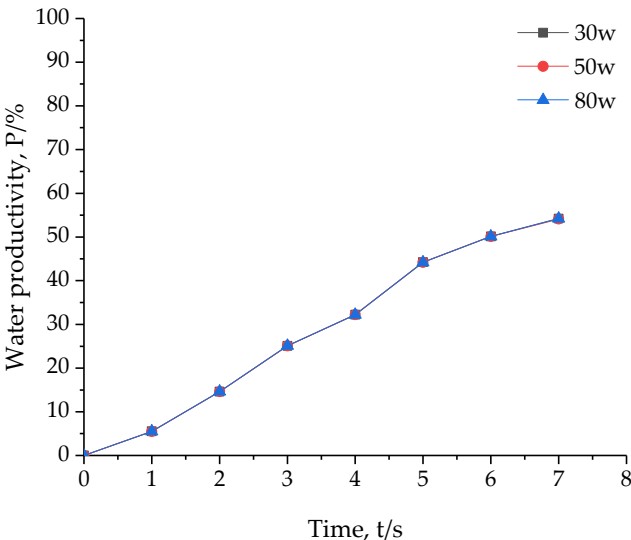

**Figure 5.** Curve of water productivity rising with time under different grid numbers.

## 3. Results

*3.1. Simulation Results and Analysis*

3.1.1. Pressure Contour Analysis

The pressure contour of section $X = 0$, when the inlet velocity is 0.6 m/s ( $Re = 19000$ ), is shown in Figure 6a. It can be seen from the figure that the pressure in the entire flow channel gradually increases stepwise from the inlet to the outlet boundary, the pressure increment is about 20 Pa, and the average pressure of the entire flow channel is about 80 Pa.

The pressure contour of section $X = 0$, when the inlet velocity was 0.6 m/s without gravity, is shown in Figure 6b. It can be seen from the figure that the pressure in the entire flow channel was evenly distributed from the inlet to the outlet boundary, the pressure increment was basically 0, and the average pressure of the entire flow channel was about 100 Pa.

The pressure contour of section $X = 0$, when the inlet velocity is 0.8 m/s ( $Re = 25000$ ), is shown in Figure 6c. It can be seen from the figure that the pressure was distributed in steps in the entire flow channel, and gradually increased from the inlet to the outlet boundary. The pressure increment was about 22 Pa, and the average pressure of the entire flow channel was about 180 Pa.

The pressure contour of the section $X = 0$, when the inlet velocity is 1 m/s ( $Re = 31000$ ), is shown in Figure 6d. As can be seen from the figure, compared to the first three working conditions, the pressure of the entire flow channel reached the maximum value, the pressure increment was basically 0, and the average pressure was about 200 Pa.

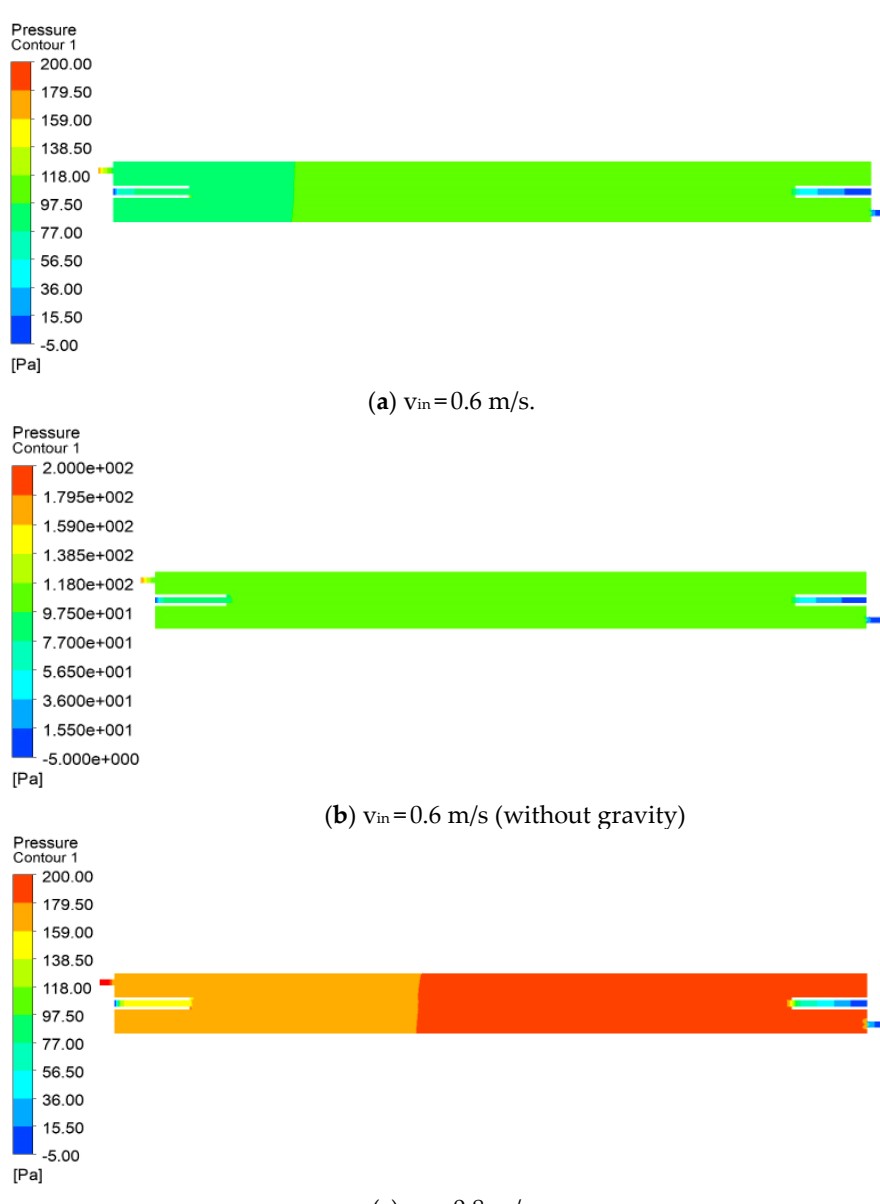

(**a**) $v_{in}$=0.6 m/s.

(**b**) $v_{in}$=0.6 m/s (without gravity)

(**c**) $v_{in}$=0.8 m/s

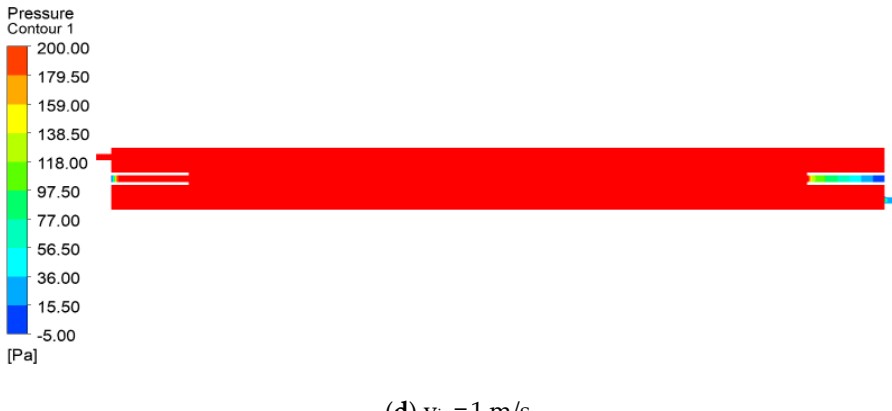

(**d**) $v_{in}$ = 1 m/s

**Figure 6.** Pressure contour on $X = 0$ section.

### 3.1.2. Velocity Contour Analysis

The velocity contour of the section $X = 0$, when the inlet velocity is 0.6 m/s, is shown in Figure 7a. It can be seen from the figure that the flow velocity of the entire flow channel presented a gradually decreasing step-like distribution from the inlet to the outlet boundary. The closer to the middle porous media zone, the smaller the flow velocity, which was due to the fact that the surface of the porous media is a tangential, no−slip boundary, the tangential velocity was 0 and the resulting normal penetration velocity. This indicates that the porous media zone can effectively reduce the tangential flow velocity.

The velocity contour of the section $X = 0$, when the inlet velocity was 0.8 m/s, is shown in Figure 7b. It can be seen from the figure that the flow velocity of the entire flow channel still presented a gradually decreasing step−like distribution from the inlet to the outlet boundary. The closer to the middle of the porous media zone, the smaller the flow velocity. Compared to the working condition at 0.6 m/s, the permeation range of the porous media zone becomes wider due to the increase in the flow velocity, which is more conducive to the increase in membrane flux.

The velocity contour of the section $X = 0$, when the inlet velocity was 1 m/s, is shown in Figure 7c. It can be seen from the figure that compared with the first two working conditions, as the flow velocity gradually increases, the permeation velocity range in the porous media zone reaches the widest range, which is beneficial to the effective improvement of membrane Flux.

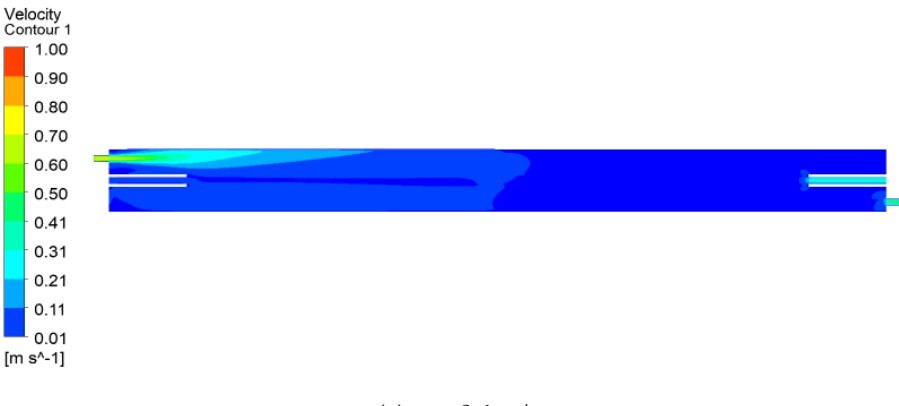

(**a**) $v_{in}$ = 0.6 m/s

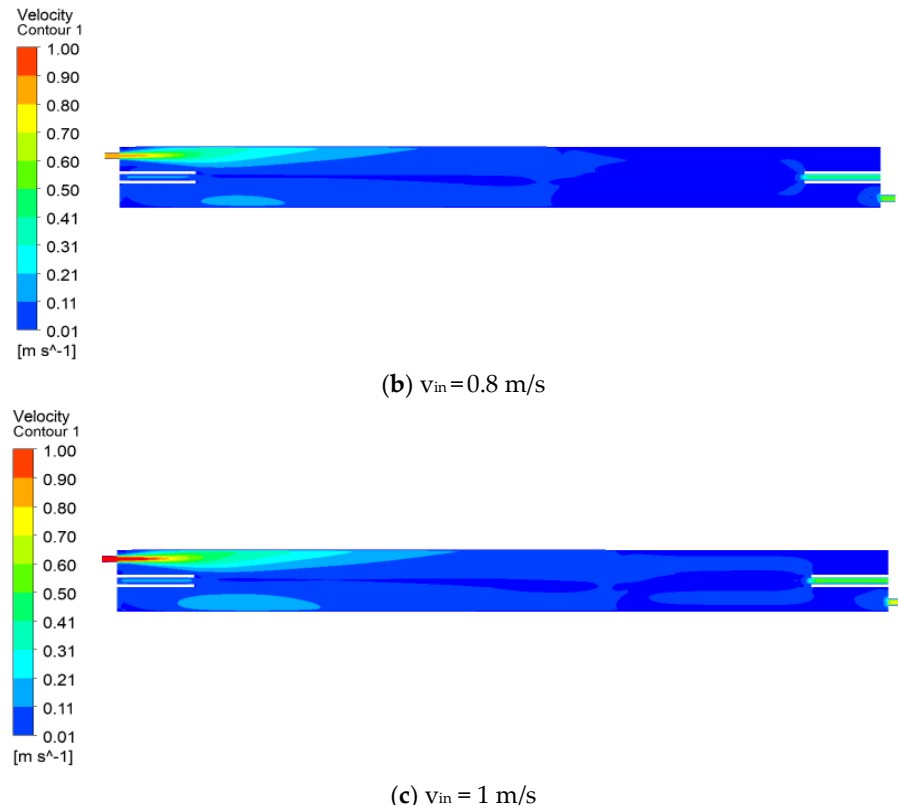

(**b**) $v_{in} = 0.8$ m/s

(**c**) $v_{in} = 1$ m/s

**Figure 7.** Velocity contour on $X = 0$ section.

### 3.1.3. Particle Concentration Contour Analysis

The particle concentration contour when the inlet velocity is 0.6 m/s, is shown in Figure 8a. It can be clearly seen from the figure that the particle concentration in the inlet zone and the upper wall surface is the highest, reaching about 0.005 mg/um$^3$. As the distance from the inlet increases, the particle concentration gradually decreases. When the particle concentration is close to the middle zone of the porous media surface, the particle concentration is the lowest, about 0.001 mg/um$^3$. The distribution is relatively uniform, while the particle concentration is higher in the porous media surface near the inlet zone, due to the suction pressure, which makes it easy to produce a large amount of accumulation.

The particle concentration contour, when the inlet velocity is 0.8 m/s, is shown in Figure 8b. It can be seen from the figure that the particle concentration in the inlet zone and the upper wall is the highest, reaching about 0.005 mg/um$^3$. With the increase in the distance from the inlet, the particle concentration gradually decreases. When approaching the middle zone of the porous media surface, the particle concentration is the lowest. Compared with the working condition of 0.6 m/s, when the flow velocity is 0.8 m/s, the particle distribution range is wider.

The particle concentration contour, when the inlet velocity is 1 m/s, is shown in Figure 8c. It can be seen from the figure that with the increase in flow velocity, the distribution range of the particle concentration reaches the widest, and the particles gradually decrease from the pipe wall to the middle of the flow channel. Compared to the first two working conditions, the particle concentration distribution is more uniform, about 0.001 mg/um$^3$. When the flow velocity is 1 m/s, the transmembrane pressure difference is the largest, as shown in Figure 6d, which is more conducive to the improvement in membrane water productivity.

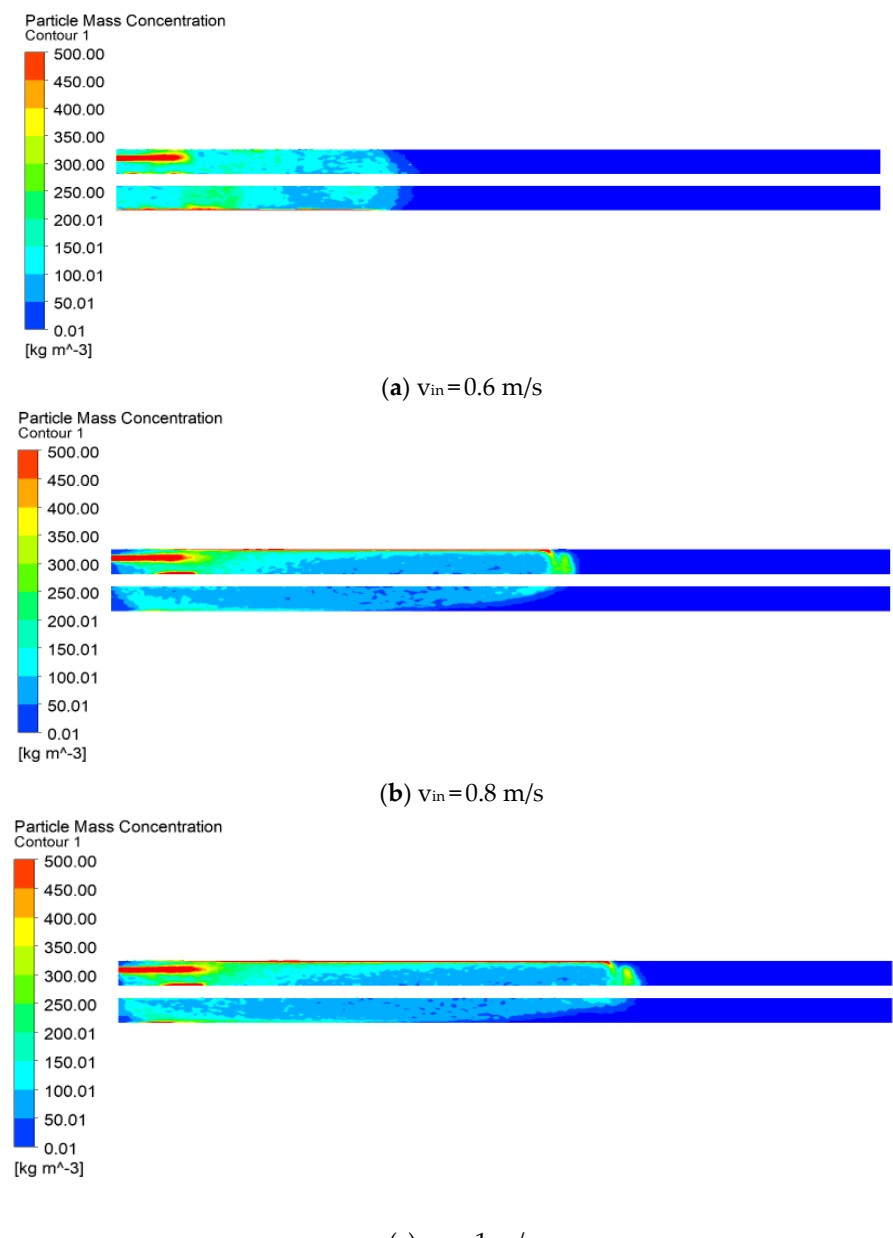

(**a**) $v_{in} = 0.6$ m/s

(**b**) $v_{in} = 0.8$ m/s

(**c**) $v_{in} = 1$ m/s

**Figure 8.** Contour of particle concentration in $X = 0$ section.

Figure 9 shows the average particle concentration in the local zone near the wall at 0.6 m/s, 0.8 m/s, and 1 m/s, respectively. Comparing the three working conditions, it can be seen that the overall particle concentration of 0.6 m/s and 0.8 m/s varies greatly, and the difference between the lowest concentration and the highest concentration is 0.0014 mg/um$^3$. However, when the flow velocity is 1 m/s, the variation range of particle concentration is relatively uniform, and the difference between the lowest concentration and the highest concentration is only 0.0005 mg/um$^3$, due to the relatively high scour velocity near the surface of the porous media zone.

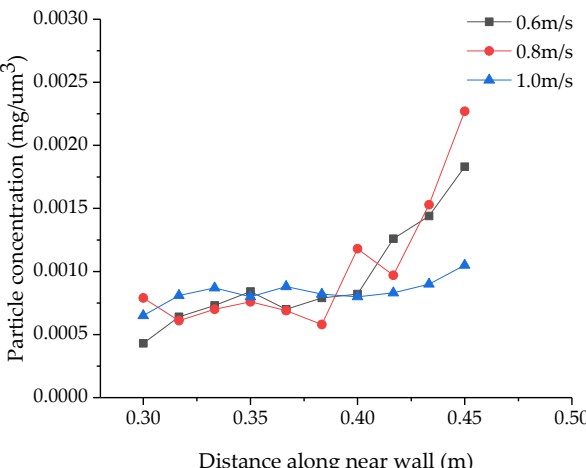

**Figure 9.** Average particle concentration near the wall.

As shown in Figure 10 $v_{in}$=0.6 m/s, when the flow velocity is 0.6 m/s, comparing the particle concentration distribution under the two working conditions with gravity and without gravity, it can be seen that when the influence of gravity is present, most of the high−concentration particles are concentrated in the lower wall zone, and only a small amount diffuses near the surface of the porous media zone. This is due to the particle's own gravity, which can generate a radial velocity. However, when there is no gravity, most particles follow the fluid along the upper wall, which is due to a radial velocity of 0 from the particle's own gravity.

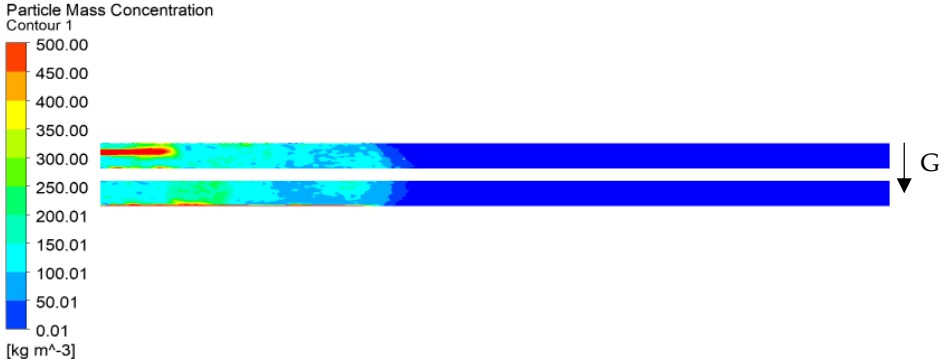

(**a**) $v_{in}$=0.6 m/s (along the Y-axis in the direction of gravity)

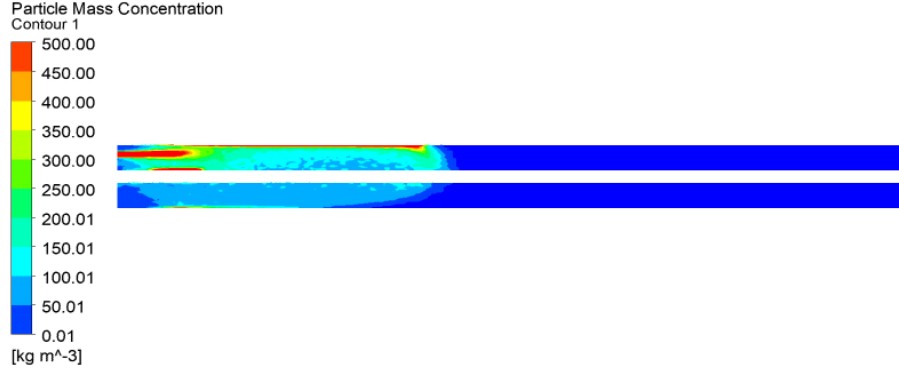

(**b**) $v_{in}$ = 0.6 m/s (without gravity)

**Figure 10.** Particle concentration contours.

The water productivity of different inlet velocities is shown in Figure 11. The results show that when the inlet velocity is 0.6 m/s, the water productivity is 45.2%; when the inlet velocity is 0.8 m/s, the water productivity is 46.1%; when the inlet velocity is 1 m/s, the water productivity is 48.5%; water productivity gradually increases with an increase in inlet velocity.

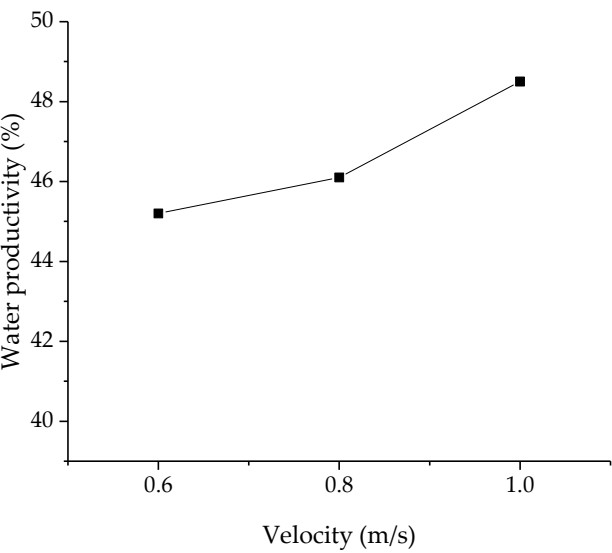

**Figure 11.** Water productivity with different inlet velocities.

## 4. Conclusions

In this paper, the micron particle aggregation during the membrane filtration process was simulated under different inlet velocities. The influence of inlet velocity on particle aggregation and water productivity was studied. The following conclusions were obtained:

(1) The porous media model can be used to simulate the flow across the semipermeable membrane, as long as accurate values for the coefficient of viscous resistance and the coefficient of inertial resistance are obtained, via experimental technology.

(2) The particle concentration distribution was affected by different Reynolds numbers. When the Reynolds number is 19,000, the particle concentration near the surface of

the porous media zone was higher, and the highest particle concentration reached 0.005 mg/um$^3$. With an increasing Reynolds number, the particle concentration near the surface of the porous media zone, gradually decreased. When the Reynolds number increased to 31,000, the particle concentration near the surface of the porous media zone was about 0.001 mg/um$^3$, which was due to the increased scour velocity near the surface. Particle deposition near the surface of the porous media zone was closely related to the Reynolds number. The larger the Reynolds number, the less particles were deposited on the surface of the porous media zone.

(3) The motion of particles in a flow field is susceptible to gravity. When there is gravity, because the particle's own gravity can generate a velocity along the direction of gravity, the particle is easy to deposit and not easy to diffuse. When there is no gravity, particles are more likely to follow the fluid motion and diffuse more freely.

(4) Comparing the three working conditions, when the Reynolds number reached 31,000, the flow velocity near the porous media zone was larger, resulting in a larger transmembrane pressure difference, which can promote higher water productivity.

This topic only discusses the effect of particle deposition on water productivity at three velocities, and the conclusions obtained have certain limitations. It is hoped that there will be more extensive discussions in the future.

**Author Contributions:** Conceptualization, P.L., Z.Z., Q.D. and B.C.; methodology, P.L.; software, X.X.; validation, X.X. and Q.W.; formal analysis, P.L., X.X. and Q.W.; investigation, X.X. and Q.W.; resources, P.L.; data curation, X.X.; writing—original draft preparation, X.X.; writing—review and editing, X.X., Q.W. and P.L.; visualization, X.X.; supervision, P.L.; project administration, P.L.; funding acquisition, P.L. All authors have read and agreed to the published version of the manuscript.

**Funding:** The present work is financially supported by the Key R&D Program of Zhejiang Province (Grant No. 2020C03081), the Joint Funds of the National Natural Science Foundation of China (Grant No. U2006221), the National Natural Science Foundation of China (Grant No. 51676173), and 521 Talents Fostering Program Funding of Zhejiang Sci-Tech University of China. The supports are gratefully acknowledged.

**Data Availability Statement:** The data that support the findings of this study are available from the corresponding author upon reasonable request.

**Conflicts of Interest:** The authors declare no conflict of interest.

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
