# Peer review of "Research on the Influence of Inlet Velocity on Micron Particles Aggregation during Membrane Filtration"

_applsci, doi:10.3390/app12157869_

Round 1

Reviewer 1 Report

Please see the attached report for comments.

Author Response

This is a well-written article and can be accepted for publication in the current

Author Response

1.The English construction needs a great deal of improvement. Eg. Line 54 – “use VOF to reduce membrane pollution” is not correct. VOF is only a method/tool. It should read more like “VOF was used to demonstrate {physical phenomenon} leading to reduction of.” Multiple instances where grammatic improvements are needed. Needs overall revision.

Reply: missing content has been added and modified in the article.

2.The Literature review is not continuous and appears disconnected. Review of Turbulent multiphase literature (Rohit Dhariwal et al) has apparently no connection to membrane review.

Reply: The documents that don’t meet the requirements have been deleted.

3.There is no justification provided for the geometric dimensions used in the simulation study. It is not clear what solver is used for simulations ? Own code? or Commercial CFD solver? The simulation methodology needs to make all assumptions very clear - isothermal, incompressible, RANS. The simulation setup fails to explain how long simulations were run for (stopping criterion?).

Reply: The geometric dimensions are based on the dimensions of the experimental model and are explained in the article; The solver and other content have been answered in detail in the article. Such as” ANSYS FULENT was used for numerical simulation, and pressure-based solver was used, the convection term of the equation adopts the second-order upwind discrete format, and the other terms of the equation adopt the central difference format. The SIMPLE algorithm is used to separate and iteratively for solving the velocity and pressure. Turbulence intensity is set to 5%. If the relative residual value of each variable is less than 10-5, which is considered that the results converge.”

4.It is felt by the reviewer that the conclusions drawn are very basic and obvious – higher velocity results in greater pressure drop (?) and lower particle concentration due to particles being carried away further (?). Also, a broader range of velocities need to be studied (including higher velocities) before drawing conclusion on an optimum velocity.

Reply: The Conclusions section has been discussed in more detail. Such as” The particle concentration distribution is affected by different Reynolds numbers. When the Reynolds number is 19000, the particle concentration near the surface of the porous media zone is higher, and the highest particle concentration reaches 0.005mg/um3. With the increasing Reynolds number, the particle concentration near the surface of the porous media zone gradually decreases. When the Reynolds number is increased to 31000, the particle concentration near the surface of the porous media zone is about 0.001 mg/um3, which is due to the increased scour velocity near the surface. Particle deposition near the surface of the porous media zone is closely related to the Reynolds number. The larger the Reynolds number, the less particles are deposited on the surface of the porous media zone.

The motion of particles in a flow field is susceptible to gravity. When there is gravity, because the particle's own gravity can generate a velocity along the direction of gravity, the particle is easy to deposit and not easy to diffuse. When there is no gravity, particles are more likely to follow the fluid motion and diffuse more freely.

Comparing the three working conditions, when the Reynolds number reaches 31000, the flow velocity near the porous media zone is larger, resulting in a larger transmembrane pressure difference, which can obtain higher water productivity”.

The paper is severely lacking in explaining/discussing the observations through physical mechanisms.

The following shortcomings are noted –

5.Use of abbreviations should be preceded by declaring them (either in nomenclature or first use). Eg. MBR, SBR, PAC

Reply: related terms have been explained in the article

6.Literature review needs to be more critical. It is not sufficient just to mention what was observed by other but try to explain why. Eg. Line 37 Ling qi et al found adding activated carbon conducive – any attempt to explain why?

Reply: Extensive revisions have been made to the introduction section as requested.

7.Figure 1 needs to give a better idea of the dimensions of inlet outlet etc.

Reply: appropriately label in the figure.

8.Boundary Conditions section mentions pressure at outlet1 and outlet2 to be -5Pa. It will help to show location of Outlet1 and Outlet2 boundaries in Figure 1 (or a separate figure)

Reply: appropriately label in the figure.

9.Figures 4-6: The scales of pressure contours for each figure is different which makes it hard to interpret and compare the pressures. A uniform scale should be used for all figures to enable comparison. Also it is not clear why the outlet pressure BC (-5Pa) is not obvious in the figures?

Reply: All contour scales have been unified and suction pressure is more pronounced.

10.Basically it is the pressure drop (difference in pressure from inlet to outlet) which is of interest. There should be discussion on pressure drop values for all cases.

Reply: Modified as requested.

11.Section 3.1.1: It is not proper to refer to YZ section unless it has been made clear in a figure where this section is located? Consider using different terminology?

Reply: Location information has been specified in different terms.

12.Figs 7-9: why are all velocities much lower than the inlet velocities (by order of magnitude!)?

Reply: all velocity contours have been modified.

13.The observations/discussion around Fig 7-9 is very weak stating the obvious. There should be a more in-depth analysis flow path, dead zones with low velocity etc.

Reply: more in-depth discussion of Velocity Contours as requested.

14.Fig 12: The particle concentration appears similar to that at 0.8 m/s (Fig 11), though the authors have commented otherwise and say that 1.0 m/s leads to improved productivity. It is not clear what is the basis of this conclusion? Needs closer examination and much better explanation.

Reply: The basis for this conclusion has been properly explained in the article. Such as” when the flow velocity is 1m/s, the transmembrane pressure difference is the largest from Figure 4, which is more conducive to the improvement of membrane water productivity.”

15.Fig 13: use consistent units discussion uses different units of particle conc. than the figure. There is no discussion on why variation in concentration is lower for the 1 m/s case?

Reply: concentration units have been unified in the figure and explained why the 1m/s concentration is smaller. Such as” when the flow velocity is 1m/s, the variation range of particle concentration is relatively uniform, and the difference between the lowest concentration and the highest concentration is only 0.0005mg/um3, which is due to the relatively high scour velocity near the surface of the porous media zone”.

16.Fig 14-15: Apparently it is the case without gravity (Fig 15) which shows wall deposition while the authors seem to suggest otherwise in the discussion.

What is direction of gravity? make it explicitly clear using a figure.

Reply: checked and adjusted own discussion and noted gravity direction in graph.

17.Fig 14-15: No attempt is made to explain the observations through physical mechanism.

Reply: it has been explained. Such as” most of the high-concentration particles are concentrated in the lower wall zone, and only a little amount diffuses near the surface of the porous media zone, which is due to the particle's own gravity can generate a radial velocity; However, when there is no gravity, most particles follow the fluid along the upper wall, which is due to radial velocity is 0 by the particle's own gravity.”

18.Fig 16: Again, no discussion at all on why productivity is improved for 1 m/s?

Reply: explained in previous chapters

Reviewer 3 Report

The paper presents an experimental and numerical analysis of fluid and particle flow through a membrane filtration system.
The authors use CFD to assess how the inlet velocity boundary condition affects the membrane filtration.

The English in the manuscript is decent, minor spell check is needed.
A mesh independence study was done.

Comments:

1. It should be specified in the abstract that the best working conditions are not general but for this specific setup.

2. It should be specified which CFD solver was used in the study.

3. It should be specified in the Governing equation section which turbulence intensity was set.

4. A reference should be added which justifies a y+ value of 0.3 for the k-Epsilon model, or rather add a reference which allows this if the Launder/Spalding standard wall function is used. Usually the k-Eps y+ requirement is a lot higher than 0.3.
Results should be written in non-dimensional form, e.g. the Reynolds number in most Figures/Tables. The Reynolds number gives a more general feel, it is hard to assess what 0.6 m/s or 0.8 m/s means.

5. The experimental results should be clearly compared to the numerical results.

6. All experimental measurement equipment should be presented for reproducibility purposes.

7. It should be stated which operating condition was used for the grid influence study.

8. Generally, the results should be more clearly presented, Figures should be compressed into single sub-Figures for a clearer overview.

Author Response

  1. It should be specified in the abstract that the best working conditions are not general but for this specific setup.

Reply: Modified in abstract

  1. It should be specified which CFD solver was used in the study.

Reply: Related content has been added. Such as” ANSYS FULENT was used for numerical simulation, and pressure-based solver was used, the convection term of the equation adopts the second-order upwind discrete format, and the other terms of the equation adopt the central difference format. The SIMPLE algorithm is used to separate and iteratively for solving the velocity and pressure. Turbulence intensity is set to 5%. If the relative residual value of each variable is less than 10-5, which is considered that the results converge.”

  1. It should be specified in the Governing equation section which turbulence intensity was set.

Reply: Related content has been added.

  1. A reference should be added which justifies a y+ value of 0.3 for the k-Epsilon model, or rather add a reference which allows this if the Launder/Spalding standard wall function is used. Usually the k-Eps y+ requirement is a lot higher than 0.3.
    Results should be written in non-dimensional form, e.g. the Reynolds number in most Figures/Tables. The Reynolds number gives a more general feel, it is hard to assess what 0.6 m/s or 0.8 m/s means.

Reply: After careful inspection, the Y+ values used in this paper do not meet the requirements of the turbulence model. Therefore, the Y+ value was modified according to the requirements of the turbulence model used.

Changed velocity to Reynolds number as requested.

  1. The experimental results should be clearly compared to the numerical results.

Reply: it needs to be explained here that the main purpose of the experimental results in the article is to determine the two resistance parameters of porous media, not to compare with the numerical simulation results.

  1. All experimental measurement equipment should be presented for reproducibility purposes.

Reply: the main equipment used in the experiment has been added.

  1. It should be stated which operating condition was used for the grid influence study.

Reply: modified as required.

  1. Generally, the results should be more clearly presented, Figures should be compressed into single sub-Figures for a clearer overview.

Reply: modified as required.

Reviewer 4 Report

My own remarks I added to attached file, 

Author Response

a) how authors took into account particles feed back on carried fluid, did they use one-way or two-way coupling approach.

Reply: In this paper, the one-way coupling method is adopted, the fluid carried by the particles can be ignored, and the effect of the fluid on the particles is mainly considered.

b) if it was two-way coupling approach how authors set particle’s feedback on carrier fluid and how the particle mass concentration has been calculated;

Reply: This article does not use the method of two-way coupling.

c) it is needed more detailed information of modeling of porous media since from where equations used in mathematical approach (8, 9) have been taken;

Reply: More detailed information has been added. Such as “Darcy's law describes the linear relationship between the seepage velocity and hydraulic gradient of water in saturated soil, also known as the linear seepage law. Porous media is modeled by the addition of a momentum source term to the standard fluid flow equations. The source term is composed of two parts: a viscous loss term and an inertial loss term.”

d) authors did not use Brownian approach for modeling of behavior of fine micron particles however the thermal Brownian random motion of particles should be taken into account.

Reply: Because the particle size used in this paper is 0.5mm, which is far larger than the nanoscale, it is greatly affected by inertia, and the Brown effect is small and can be ignored.

Round 2

Reviewer 2 Report

Thanks for the revisions. I have attached and highlighted my comments.

Author Response

1.Line 229, 233, 241,333 (check other places): Fig number is missing

Reply: completed

2.Please correct the typo “FLUENT”

Reply: Corrected.

3.Please report pressure drop in section 3.1.1

Reply: has been added to the text.

4.It would be beneficial if the authors added a few lines about future work at the end of the paper. For eg. any plans to increase the coverage of velocities studied in the future?

Reply: Added at the end of the text. Such as “This topic only discusses the effect of particle deposition on water productivity at three velocities, and the conclusions obtained have certain limitations, and it is hoped that there will be more extensive discussions in the future.”

Reviewer 3 Report

The authors have made some changes to improve the manuscript, however there exists a major issue. When using a different/adequate grid y+ value (min 30) for the k-Epsilon model, the results should obviously be at least slightly different. It is apparent through the mesh independence procedure, particle deposition results and even the mesh schematic which remains unchanged, that the results are identical to those of a smaller y+ value (0.3). A rigorous numerical validation is essential for studies where there is no comparison with experimental data.

Author Response

The authors have made some changes to improve the manuscript, however there exists a major issue. When using a different/adequate grid y+ value (min 30) for the k-Epsilon model, the results should obviously be at least slightly different. It is apparent through the mesh independence procedure, particle deposition results and even the mesh schematic which remains unchanged, that the results are identical to those of a smaller y+ value (0.3). A rigorous numerical validation is essential for studies where there is no comparison with experimental data.

Reply: Sorry, Y+=0.3 is a typo, I wrote the value in the grid skewness in the previous row here, Y+ was originally 30.

Reviewer 4 Report

It would be interested to see the method for evaluation of the particle mass concentration used in paper. The point is to see transformation from Lagrangian to Euler approach for calculation of the particle mass concentration. 

Author Response

It would be interested to see the method for evaluation of the particle mass concentration used in paper. The point is to see transformation from Lagrangian to Euler approach for calculation of the particle mass concentration.

Reply: In this paper, the Euler-Lagrange method is used to calculate the mass concentration of particles. The fluid phase is treated as a continuous phase, and the time-averaged Navier-Stokes equation is directly solved, while the discrete phase is obtained by calculating the motion of a large number of particles in the flow field. There will be an exchange of momentum, mass and energy between the discrete phase and the fluid phase.

This manuscript is a resubmission of an earlier submission. The following is a list of the peer review reports and author responses from that submission.

Round 1

Reviewer 1 Report

Unfortunately, this paper doesn't represent a serious research work.

Author Response

The review comments mentioned below have been carefully revised.

Reviewer 2 Report

Please see the attached report for comments.

Reviewer 3 Report

The topic is interesting and the selected method to simulate the phenomenon is quite appropriate. There are a few typos and also writing errors that are not according to standard academic writing, e.g., using personal pronouns, wrong style of citations within the text, etc. 

The introduction must be improved to include more recent studies in the field. Below, you can find more specific comments.

Line 100: The k-e equations are well-known. Maybe you don’t need to mention equations here unless you need to highlight a specific parameter within it.

Line 101: What do the authors mean by this sentence: ‘which is an equation about the turbulent energy dissipation rate introduced on the basis of the single-equation model.’ Please amend this sentence.

Line 122: More information concerning the mesh quality is required here, e.g., the skewness of the mesh, the resolution near the wall. Also, please highlight how you have captured turbulent sub-layer that affect the accuracy of your calculation.

Not enough information is provided about the data used for this study. Please provide more information about the case study and its specifications.

The presentation of the results are good but your conclusion can be improved when the above-mentioned comments are applied to provide some arguments about how your selected mesh affected the solutions. 

I hope my comment can help to improve the quality of your nice work. 

Round 2

Reviewer 2 Report

Authors have satisfactorily addressed my comments, but two studies have been incorrectly referenced.

In Ref.[24] order of authors name has been swapped and also the authors names are incomplete. It should be:

Dhariwal, R.; Bragg, A. D. Enhanced and suppressed multiscale dispersion of bidisperse inertial particles due to gravity. J.Physical Review Fluids, 2019, 4(3), 034302

At the end of Ref. [28], it should be "205–249" instead of "205-409". Also, name of last co-author "Koch, D. L." is missing for this reference.

Author Response

Authors have satisfactorily addressed my comments, but two studies have been incorrectly referenced.

In Ref.[24] order of authors name has been swapped and also the authors names are incomplete. It should be:

Dhariwal, R.; Bragg, A. D. Enhanced and suppressed multiscale dispersion of bidisperse inertial particles due to gravity. J.Physical Review Fluids, 2019, 4(3), 034302

At the end of Ref. [28], it should be "205–249" instead of "205-409". Also, name of last co-author "Koch, D. L." is missing for this reference.

Modified in the text

Reviewer 3 Report

Thanks for the authors' effort in updating the manuscript and submitting the new version. Most of my comments have been applied. Just, I don't see any information about capturing the near-wall zone. Authors must clarify how they have captured this. Have you used a wall function? How the mesh is adjusted near the wall? 

Still some typos and errors exist in the English writting. Please review the manuscript throughout. 

Author Response

 I don't see any information about capturing the near-wall zone. Authors must clarify how they have captured this. Have you used a wall function? How the mesh is adjusted near the wall? 

Added in the text

Still some typos and errors exist in the English writting. Please review the manuscript throughout. 

Has been carefully checked and modified

Round 3

Reviewer 3 Report

The provided information is not satisfactory and it doesn't support the arguments and justifications appropriately. I think the paper must address the key information about a CFD simulation and mesh construction by presenting more details. 

Author Response

The provided information is not satisfactory and it doesn't support the arguments and justifications appropriately. I think the paper must address the key information about a CFD simulation and mesh construction by presenting more details. 

Modified as required